# Current Knowledge and Perceptions of Bariatric Surgery among Diabetologists and Internists in Poland

**DOI:** 10.3390/jcm11072028

**Published:** 2022-04-05

**Authors:** Karolina Zawadzka, Krzysztof Więckowski, Tomasz Stefura, Piotr Major, Magdalena Szopa

**Affiliations:** 12nd Department of General Surgery, Faculty of Medicine, Jagiellonian University Medical College, 30-688 Krakow, Poland; karolina.zawadzka@onet.pl (K.Z.); k.wieckowski@student.uj.edu.pl (K.W.); tomasz.stefura@gmail.com (T.S.); piotr.major@uj.edu.pl (P.M.); 2Centre for Research, Training and Innovation Jagiellonian (CERTAIN Surgery), 30-688 Krakow, Poland; 3Department of Metabolic Diseases, Faculty of Medicine, Jagiellonian University Medical College, 30-688 Krakow, Poland

**Keywords:** bariatric surgery, diabetologists, internists, obesity, survey

## Abstract

Perioperative care and follow-up after bariatric surgery (BS) engage various medical professionals. It is key for them to be well informed about these procedures. However, knowledge and attitudes may be not satisfactory enough to provide proper care. We aimed to assess knowledge and perceptions of BS among diabetologists and internists. A total of 34 diabetologists and 30 internists completed the electronic questionnaire. There were no differences in self-estimated knowledge between them, except regarding items related to the treatment of diabetes and metabolic control. Several misconceptions were identified in the questions testing the understanding of key issues in BS. Most participants considered BS effective in weight loss and metabolic control. A total of 75% highlighted the lack of appropriate equipment for dealing with morbidly obese patients. Interestingly, in a multivariable linear regression model, self-estimated knowledge was the only variable associated with frequency of referrals to bariatric surgeons. A total of 92% of respondents were interested in broadening their knowledge. Guidelines for long-term follow-up and funding were the most frequently chosen topics to explore. The study showed a positive attitude of diabetologists and internists towards surgical treatment of obesity and identified some significant gaps in knowledge. The results may be helpful in planning trainings to provide the best care for patients suffering from morbid obesity.

## 1. Introduction

Obesity is a growing challenge for healthcare systems worldwide, and its prevalence has increased in recent years [1]. Additionally, actions taken against the COVID-19 pandemic, especially national lockdowns, may worsen the problem even more [2,3].

Non-surgical treatments of obesity such as dietary and physical activity intervention, behavioral therapy, and even pharmacotherapy are not sufficient in some patients, particularly in the morbidly obese, and a surgical approach may be more effective [4]. It must be highlighted that bariatric surgery (BS) is the beginning of extensive treatment involving crucial postoperative care rather than an immediate solution of this problem. Management of patients undergoing BS includes numerous challenges, especially related to the many comorbidities seen in this population. In response to this, all recommendations emphasize the importance of interdisciplinary care [5,6,7], and such treatments increasingly engage more specialists in various fields of medicine.

So far, surveys evaluating the knowledge and attitudes to BS were conducted among many groups of medical professionals, including general practitioners as well as secondary healthcare providers, and they showed that understanding of this topic is insufficient [8,9,10]. Nonetheless, to our knowledge, no study has focused on diabetologists and internists. There is abundant evidence that a substantial majority of obese patients with diabetes experience complete resolution or improvement after BS [11]. Thus, we believe that awareness of the benefits and effectiveness of weight loss surgery among specialists involved in the treatment of diabetes is extremely important.

The aim of this research was to assess the knowledge and perceptions of BS among diabetologists and internists and to identify areas for improvement to provide better care to patients suffering from obesity. The results of this study could help organizers of future training courses and conferences in the selection of topics to be addressed.

## 2. Materials and Methods

This was a cross-sectional study carried out via an online, anonymous, self-administered structured questionnaire. The survey was designed using Google Forms. The study population included practicing internists, diabetologists, internal medicine trainees, and diabetology trainees attending two large virtual conferences: “Advances in Diabetology”, Krakow, Poland in October 2019, and “Bariatric Surgery without Borders” in January 2021. An invitation to complete the survey was also published on a well-known portal for Polish doctors interested in the field of diabetology (diabetologia.mp.pl) (accessed on 13 March 2022). Specialists and trainees in specializations other than internal medicine or diabetology, officials, and dieticians were excluded from this study. This study followed the guidance of the Checklist for Reporting Results of Internet E-Surveys (CHERRIES) guidelines [12].

The questionnaire was developed in consultation with a group of experts on BS and diabetology. To check comprehension and clarity of the questions, the question stem was submitted in a pilot test to doctors working in the Department of Metabolic Diseases prior to the utilization of the questionnaire. The online survey provided a brief description of the study, eligibility requirements, procedures, and instructions on how to answer the questionnaire. The questionnaire consisted of four sections with 32 questions total. Answers to all survey questions were mandatory. The survey is presented in Appendix A.

In the first part of the survey, sociodemographic variables were collected using five open-ended questions (age, years of clinical practice, number of morbidly obese patients seen monthly, and number of morbidly obese patients with diabetes seen per month) and four closed single-choice questions (gender, specialty, type of practice setting, number of patients seen monthly). Additionally, respondents were asked to indicate how often they refer morbidly obese patients to a surgical consultation for BS. The 1–5 rating scale was used, where 1 meant *I never refer to consultation* and 5—*I refer each morbidly obese patient to consultation.*

In the second section, we asked physicians to rate their level of knowledge about BS on a five-point scale (1—*I know nothing about it,* 5—*I have an excellent knowledge about it*). Specifically, the questions investigated the rules of qualification for the surgical treatment of obesity, preparing patients for BS, principles and scope of multidisciplinary teams caring for bariatric patients, knowledge about the basic types of surgical procedures in the field of BS and their mechanisms, knowledge about long-term consequences and possible health problems in a group of patients after BS, and knowledge about the principles of diabetes treatment in patients admitted to hospital for BS. Next, we asked about doctors’ attitudes and beliefs regarding the efficacy and value of BS using a five-point Likert scale: strongly disagree, disagree, I do not know/I have no opinion, agree, and strongly agree. Questions regarded the availability of appropriate tools for managing morbidly obese patients, perception about the effectiveness of BS in the treatment of diabetes, and its impact on glycemic control as well as opinions on the need for vitamin and mineral supplementation after weight loss surgery.

The third section was designed to test respondents’ knowledge of bariatrics. We used both single-choice and multiple-choice questions to check physicians’ familiarity with the eligibility criteria for BS, familiarity with contraindications for BS, principles of pre- and postoperative care of bariatric patients, and knowledge about the most commonly performed BS in Poland. Questions were created based on Polish recommendations in the field of BS [13]. In multiple-choice questions, we considered the answer valid only if the respondent indicated all the correct options. Correct responses were summed up to obtain a total knowledge score for each participant. We also compared whether there was a difference in the level of knowledge between physicians with and without specialization in diabetes.

Finally, we asked respondents whether they would be interested in broadening their knowledge of BS. In the last question, the providers could indicate issues about BS they would be most interested in. In addition, respondents could also add their own answer to this question.

Participant confidentiality and anonymity was maintained all through the process of the research. No incentives were offered to participants and attendance was voluntary. Participants were informed about the aim of the study, and informed consent was obtained electronically prior to the beginning of the survey. The study was performed in accordance with the ethical standards laid down in the 1964 Declaration of Helsinki and its later amendments. The study was approved by the Bioethics Committee of the Jagiellonian University (1072.6120.20.2020).

STATISTICA v.13 (StatSoft Inc., Tulsa, OK, USA) was used for statistical analysis. Categorical data were analyzed using chi-squared tests with or without corrections and presented as the number of subjects (*n*) and percentage of total data (%). Normality of the data was tested with Shapiro–Wilk test. The quantitative data we collected were non-parametric, so we compared them using the two-tailed Mann–Whitney test. Continuous data are presented as median and interquartile range.

Furthermore, frequency of recommending BS for morbidly obese patients was used as the dependent variable in a linear regression model. The model contained the following independent variables: respondents’ age and gender, specialty, time in practice, workplace, median number of morbidly obese patients seen per month, median number of morbidly obese patients with diabetes seen per month, knowledge about bariatric surgery expressed by the number of correct answers to questions about BS, self-estimated knowledge. All variables were included in univariate as well as multivariable linear regression models. Results were considered statistically significant when the *p*-value was found to be less than 0.05.

## 3. Results

One hundred and ten participants attended the diabetes conference and one hundred and thirty-five people attended the bariatric conference in 2021, respectively. A total of 34 participants during the second conference declared that they had already completed the questionnaire during the first conference. Of the two hundred and eleven participants, eighty participants in both conferences completed the questionnaires, which gave a 37.9% response rate. Sixteen questionnaires were completed by nutritionists and doctors of other specialties, so we excluded those data. In total, we obtained 64 responses from internists and diabetologists. Three-quarters of the respondents were women. The majority of respondents—41 (64%)—were internal medicine specialists, and nearly half of doctors (46.9%) self-identified as diabetes specialists. Twenty-four (80%) of the diabetologists also had a specialization in internal medicine. A total of 11.8% of doctors qualified as diabetologists, and 40% of internists were undergoing specialization training. Most of the respondents had more than 200 patients a month. Demographics and practice characteristics of responders are summarized in Table 1.

In the first section, 56% of participants rated their own knowledge regarding treatment of diabetes in a patient admitted to hospital for bariatric surgery as quite good or excellent. The participants also stated that they were familiar with the rules of qualification for the surgical treatment of obesity—30% of them rated their level of knowledge at 4 and 16% at 5, on a five-point scale. With regard to knowledge about basic types of BS procedures and their mechanisms, as well as long-term consequences and possible health problems in patients after BS, most of the participants rated their knowledge at 3 or 4 points. In contrast, half of the respondents assessed their knowledge of the goals, scope, and how these multidisciplinary teams care for bariatric patients at only 1 or 2 points (Figure 1). Self-assessed knowledge of BS did not differ between diabetologists and non-diabetologists, except for the principles of diabetes management in patients admitted to hospital for BS (median 3 points for non-diabetologists and 4 points for diabetologists, *p* = 0.012).

Four statements addressed respondents’ attitudes and perceptions about bariatric care. Three-quarters of respondents disagreed with the statement “I have access to the appropriate tools to deal with morbidly obese patients (e.g., scales >150 kg, beds, diagnostic equipment)” and only 13% replied affirmatively. The vast majority of physicians, 81.3%, claimed that vitamins and microelements should be supplemented in all patients after bariatric surgery. Similarly, 90.6% agreed with the statement that bariatric surgery is an effective treatment for metabolic syndrome. When asked whether “Bariatric surgery has a better effect on glycemic control than an intensive conservative treatment?”, 84.4% of respondents replied affirmatively. Details are presented in Figure 2.

Table 2 summarizes current diabetologist and internist knowledge and awareness of BS. A majority of respondents were able to correctly identify the eligibility of patients under 18 for BS, most frequently performed BS in Poland, as well as the recommended scheme for outpatient follow-up after BS. In contrast, 35.9% of respondents correctly identified rules concerning the eligibility of patients for bariatric surgery, and only every fifth health provider had comprehensive knowledge about the need for contraception after BS. Notably, none of the respondents correctly recognized all absolute contraindications for BS in adult patients and criteria for the resolution of type 2 diabetes and comorbidities in a patient after BS who discontinued pharmacotherapy. In the multiple-choice questions about absolute contraindications for BS and criteria for the resolution of type 2 diabetes, the median number of correctly indicated contraindications was three out of the six correct ones given in the questions.

Diabetologists were more likely to correctly identify which metabolic control criteria are an indication for postponing a scheduled bariatric procedure than internists (45.3 vs. 31.3%, *p* = 0.02). As anticipated, diabetologists were more likely to know what blood glucose levels should be maintained in the perioperative period than their non-diabetologist counterparts (94.1 vs. 60%, *p* = 0.001). Likewise, diabetologists more often gave the correct answer regarding the perioperative mortality after BS (85.3 vs. 56.6%, *p* = 0.01).

In our survey, we also asked respondents to identify the three most important additional tests necessary to be performed before BS. The top three reasons indicated by providers included coagulation parameters (activated partial thromboplastin time, prothrombin time presented as international normalized ratio, bleeding time), blood count test, and serum creatinine concentration (92.2%, 84.4%, and 50% of respondents, respectively). None of the respondents indicated resting electrocardiogram or glycated hemoglobin concentration as one of the most important tests before surgery.

The linear regression model with the item “How often do you refer patients with morbid obesity to a surgical consultation for bariatric surgery?” as the dependent variable is shown in Table 3. Diabetologists declared more frequent referral of patients to surgical consultations for BS than non-diabetologists (b = 0.265, *p* = 0.034). Of note, self-estimated knowledge was also significantly associated with the greater frequency of recommending bariatric procedures in morbidly obese patients (b = 0.386, *p* = 0.0016). In the multivariate analysis, level of self-estimated knowledge remained an independent factor linked with the frequency of referring patients to a bariatric consultation.

Overall, 59 (92.2%) respondents were interested in broadening their knowledge about bariatrics. Table 4 presents the participants’ expectations on the need to expand their knowledge of BS. The main suggestion indicated by the physicians was the need for guidelines for long-term follow-up of patients after BS (89.1%). The second most frequently mentioned area was the reimbursement policy for BS (78.1%). Over half of them also wanted to know the rules of qualifying for BS and location of bariatric surgery centers (57.8% and 53.1%, respectively).

## 4. Discussion

To the best of our knowledge, this is the first study that provides deep insight into the awareness, attitudes, and experiences of surgical obesity treatment among diabetologists and internists. Our results show that most healthcare providers were knowledgeable on the crucial role of BS and its effectiveness, not only in weight loss, but also in maintaining adequate glycemic control in patients with diabetes and in treating metabolic syndrome. However, the majority of them lacked a full understanding of BS, especially about eligibility criteria for BS, biochemical metabolic control criteria for postponing elective BS, and the need for postoperative contraception. Furthermore, none of the study participants were able to correctly identify all contraindications for BS and criteria for the resolution of type 2 diabetes and other comorbidities in a patient after BS. Study respondents also pointed out meeting the special equipment needs of extremely obese patients (scale >150 kg, beds, diagnostic equipment) as a significant barrier to ensuring adequate care for obese patients.

In recent years, a growing body of literature has studied general practitioner (GP) knowledge and attitudes towards BS [9,10,14,15,16,17,18,19,20]. Notably, few primary care providers reported referring eligible patients for a bariatric consultation. Non-surgical approaches such as dietary intervention, increased physical activity, and behavioral therapy were still seen as the most effective and were therefore recommended by GPs much more often [17,18]. Although GPs play an important role in the care of obese patients, secondary healthcare specialists are usually responsible for the specialized treatment of obesity-related conditions. It must be highlighted that BS, beyond its profound weight loss effects, sustains improvements in metabolic dysfunction secondary to obesity [21,22]. Specifically, there is unequivocal evidence that BS offers a safe and more effective alternative to intensive conservative treatment for achieving glycemic control in obese diabetic patients [23]. Interestingly, BS could also induce both short- and long-term diabetes remission [24,25]. Our respondents had a largely positive attitude to BS and supported the belief that BS is a valid treatment strategy for obese patients with both type 2 diabetes mellitus and metabolic syndrome. Notwithstanding, none of the respondents knew all the criteria for the resolution of type 2 diabetes and comorbidities in a patient after BS who discontinued pharmacotherapy.

In our study, healthcare professionals reported having a decent knowledge of diabetes management in a patient admitted to hospital for BS. As might be expected, diabetologists rated their level of knowledge on this subject higher than non-diabetologists. Consequently, diabetes specialists significantly more often than internists correctly indicated what blood glucose levels should be maintained in the perioperative period. The survey also identified that diabetes specialists were more likely to correctly identify what biochemical metabolic control criteria were an indication for postponing planned BS. There are no studies on the knowledge and approach of diabetologists to bariatrics, albeit several authors have compared endocrinologists with other specialists or family doctors [8,26,27,28]. These studies have focused almost exclusively on referral patterns, and the results are inconsistent. While some authors noted that endocrinologists were more familiar with the eligibility criteria for BS and referred patients to BS more often than other specialists [8,26,27], others claimed that endocrinologists were the least willing group of specialists to address surgical weight-loss options in obese patients [28].

One of the main barriers to referral to BS is a concern about the postoperative complications [20,26,29]. Perlman et al. found that the most common reason for not referring patients to BS was the perception of BS as having a high risk of complications and death [30]. Concomitantly, Lopez et al. showed that only slightly more than half of GPs correctly identified the mortality rate of BS as <1% [18]. These findings are consistent with the responses of the surveyed internists—56.7% of them correctly pointed out the mortality rate of BS. In contrast, almost three-quarters of diabetologists demonstrated adequate knowledge on this matter.

Stolberg et al. and Conaty et al. revealed that there is limited understanding of the eligibility criteria for BS among GPs [14,20]. This is in line with our findings showing that only every third doctor knew the exact criteria for qualifying patients for BS. In addition, none of the respondents were able to correctly identify contraindications for BS. A knowledge gap in understanding eligibility criteria may contribute to bariatric surgery not being offered to the patients who would benefit most from it [31]. Women who have undergone bariatric surgery are advised to delay conception for at least 12 to 24 months [32]. Thus, some studies investigated awareness of the need to use contraception in female patients and revealed tremendous misconceptions in this area. Only 20.3% of physicians had adequate knowledge about the duration of contraception after BS. These results share several similarities with the findings of Ben-Porat et al. [33]. They reported that only a minority of bariatric surgeons recommended postoperative contraception or referred women to contraceptive advice. Moreover, most practitioners also reported a lack of accurate knowledge of contraception after BS.

Access to the appropriate equipment and resources for the treatment of obese patients remains a challenge. In our study, three-quarters of respondents either disagreed or strongly disagreed with the statement that they have adequate access to the tools to care for obese patients. These findings are complementary to a study of Auspitz et al., declaring that only 32.1% of health providers had appropriate equipment for managing obese patients [10]. Likewise, Ferrante et al. found that almost half of primary health practitioners did not have scales that could be used to weigh morbidly obese patients [34]. It has been suggested that knowledge towards weight loss surgery is related to the frequency of BS referrals [17]. Interestingly, we found no correlation between the number of correctly answered questions regarding the most important BS issues and the frequency of referring patients to BS. Nevertheless, the higher the self-estimated knowledge level, the more often doctors declared referring eligible patients to a bariatric consultation. Hence, it is essential to reduce stigma on BS and increase awareness of its safety and effectiveness. Fortunately, 92.2% of physicians were enthusiastic about broadening their knowledge about bariatrics. In order to plan possible future training about BS, we asked doctors what their main expectations were for such courses. Nearly 90% of respondents admitted that they were most interested in learning the guidelines for long-term follow-up of patients after BS. The vast majority of doctors also expressed their willingness to get acquainted with the principles of financing the BS by the National Health Fund. At this point, we would like to emphasize that in Poland, access to all bariatric surgeries could be reimbursed, so the financial aspect should not be a barrier to access of these procedures [35].

There are several strengths and limitations of this study that need consideration. A strength was the inclusion of both diabetologists and internists. Diabetologists are specialists who encounter numerous metabolic effects of obesity on a daily basis, and their knowledge and approach to bariatrics seems to be extremely important for the effective treatment of such patients. Another innovative aspect of our research is that we not only asked respondents to subjectively evaluate their knowledge and approach to bariatrics but also tested their knowledge with an objective questionnaire. We also examined physician expectations of BS training in order to plan future courses according to their needs. The findings of this study must be seen in light of some limitations. Firstly, the overall number of 64 respondents is relatively low. It should be noted that, unlike the questionnaires focusing on family physicians, who are the most numerous professional group among physicians, we chose a narrow group of diabetes specialists as the target population for the study. Research conducted in the USA with secondary healthcare providers included 10, 14, and 81 endocrinologists, respectively, which is comparable with ours [26,27,28]. Some concerns may have been raised over the generalizability of these results, given our 37.9% response rate. However, this value was found to be typical for similar online surveys, where the response rate ranged from 12.4% to 44% [8,10,14,15,18,26]. Secondly, we cannot rule out a selection bias, as healthcare providers with a more positive attitude towards BS may be more willing to participate in this type of study. The third limitation concerns the frequency of referrals, which was assessed by physicians on a subjective scale from 1 to 5, and we had no data on the real number of such referrals. Moreover, the questionnaire had not been validated. The questions were based on the clinical knowledge of a group of experts in bariatric surgery and diabetes and a critical review of the literature. The questions were also presented in a pilot survey to doctors working in the Department of Metabolic Diseases before the questionnaire was applied.

## 5. Conclusions

While the majority of diabetologists and internists consider bariatric surgery as a crucial tool for managing the detrimental health effects of obesity, many misconceptions about bariatric surgery still exist. We demonstrated that diabetologists knew significantly more about the metabolic criteria for postponement of surgical treatment of obesity and the management of diabetes in bariatric patients and that they were more familiar with the safety of these surgeries. Nevertheless, we have shown that a significant lack of knowledge in many aspects of bariatric surgery exists in both diabetologists and internists. We hope that findings from our study will contribute to the planning of extensive training of physicians in bariatric surgery and will thus bridge the gap between studies emphasizing effectiveness of bariatrics in treating metabolic complications of obesity and the application of this knowledge in everyday practice.

## Figures and Tables

**Figure 1 jcm-11-02028-f001:**
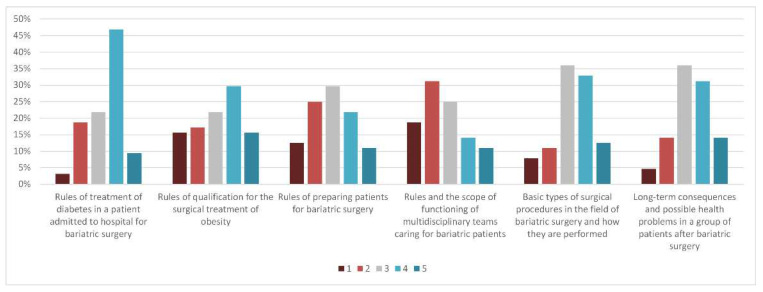
Frequency distributions in percentage in reply to questions assessing self-estimated knowledge about bariatrics.

**Figure 2 jcm-11-02028-f002:**
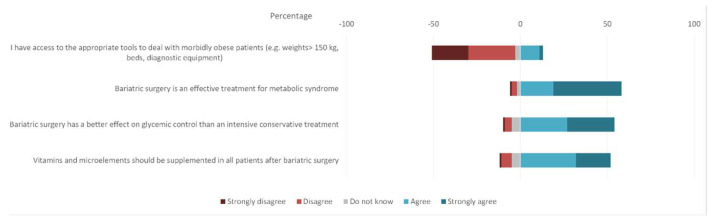
Respondents’ attitudes and perceptions about bariatric care.

**Table 1 jcm-11-02028-t001:** Demographic characteristics of respondents.

Characteristic		Value
Median age, years (IQR *)		45 (38.0–52.5)
Gender (female), *n* (%)		48 (75.0)
Specialty	Diabetologists, specialists, *n* (%)	30 (46.9)
	Diabetologists, trainees, *n* (%)	4 (6.3)
	Internists, specialists, *n* (%)	41 (64.0)
	Internists, trainees, *n* (%)	12 (18.8)
Practice setting	Outpatient specialist care, *n* (%)	28 (43.8)
	Primary healthcare, *n* (%)	24 (37.5)
	University-affiliated hospital, *n* (%)	15 (23.4)
	Non-university-affiliated hospital, *n* (%)	18 (28.1)
Median time in practice, years (IQR)		20 (10–27)
Number of patients seen per month, *n* (%)	<50	17 (26.6)
	50–100	10 (15.6)
	100–150	8 (12.5)
	150–200	11 (17.2)
	>200	18 (28.1)
Median number of morbidly obese patients seen per month, *n* (IQR)		9 (2–15)
Median number of morbidly obese patients with diabetes seen per month, *n* (IQR)		5 (2–10)

* IQR—interquartile range.

**Table 2 jcm-11-02028-t002:** Knowledge and awareness of bariatric surgery.

Area of Knowledge	All Physicians (*n* = 64)	Diabetologists ^1^ (*n* = 34)	Non-Diabetologists ^2^ (*n* = 30)	*p*-Value
Correct identification of the rules concerning the eligibility of patients for bariatric surgery, *n* (%) ^3^	23 (35.9)	11 (32.4)	12 (40)	0.53
Correct identification of absolute contraindications for bariatric surgery in adult patients, *n* (%) ^3^	0 (0)	0 (0)	0 (0)	NA
Knowledge about the eligibility of patients under 18 for bariatric surgery, *n* (%)	44 (68.8)	21 (61.8)	13 (43.3)	0.14
Knowledge of the most frequently performed bariatric surgery in Poland, *n* (%)	42 (65.6)	23 (67.7)	19 (63.3)	0.72
Knowledge of what biochemical metabolic control criteria are an indication for postponing a scheduled bariatric surgery, *n* (%) ^3^	29 (45.3)	20 (58.8)	9 (30)	0.02
Knowledge of what blood glucose levels should be maintained in the perioperative period, *n* (%)	50 (78.1)	32 (94.1)	18 (60)	0.001
Correct identification of 30-day mortality risk after bariatric surgery, *n* (%)	46 (71.9)	29 (85.3)	17 (56.7)	0.01
Knowledge of the recommended scheme for outpatient follow-up after bariatric surgery, *n* (%)	44 (68.8)	25 (73.5)	19 (63.3)	0.38
Knowledge of the criteria for the resolution of type 2 diabetes and comorbidities in a patient after bariatric surgery who discontinued pharmacotherapy, *n* (%) ^3^	0 (0)	0 (0)	0 (0)	NA
Knowledge of the need to use contraception in women after bariatric surgery, *n* (%)	13 (20.3)	9 (26.5)	4 (13.3)	0.19

^1^ Diabetology specialists or during specialization in diabetology. ^2^ Internal medicine specialists or during specialization in internal medicine. ^3^ Multiple-choice questions.

**Table 3 jcm-11-02028-t003:** Linear regression model with “How often do you refer patients with morbid obesity to a surgical consultation for bariatric surgery?” as the dependent variable.

Independent Variables	Univariate Analysis		Multivariate Analysis	
	Beta-Coefficients	*p*-Value	Beta-Coefficients	*p*-Value
Age (years)	0.073	0.57	0.095	0.88
Female (yes vs. no)	0.114	0.37	−0.081	0.54
During/after specialization in diabetology (yes vs. no)	0.265	0.03	−0.080	0.60
Time in practice (years)	0.063	0.62	−0.243	0.69
Work in university-affiliated hospital (yes vs. no)	−0.156	0.22	0.238	0.09
Median number of morbidly obese patients seen per month	0.128	0.31	−0.117	0.68
Median number of morbidly obese patients with diabetes seen per month	0.218	0.09	0.242	0.41
Number of correct answers to questions about bariatric surgery	0.079	0.54	−0.104	0.48
Self-estimated knowledge (number of points)	0.386	0.002	0.429	0.002

**Table 4 jcm-11-02028-t004:** Healthcare providers’ expectations regarding the need to broaden their knowledge of bariatric surgery.

Expectations	*n* (%)
Guidelines for long-term follow-up of patients after bariatric surgery, *n* (%)	57 (89.1)
Principles of reimbursement of bariatric surgeries by the National Health Fund, *n* (%)	50 (78.1)
Rules of qualifying patients for bariatric surgery, *n* (%)	37 (57.8)
Location of bariatric surgery centers, *n* (%)	34 (53.1)
The effectiveness of bariatric surgeries, *n* (%)	27 (42.2)
Knowledge of the types of bariatric surgeries, *n* (%)	20 (31.3)
Dietary management after surgery, *n* (%) ^1^	2 (3.1)
Cooperation of a primary care physician with a team of bariatric specialists, *n* (%) ^1^	1 (1.6)

^1^ Expectations suggested by respondents in the “other answer” option.

## Data Availability

The data presented in this study are available on request from the corresponding author.

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
