# Peer review of "Current Knowledge and Perceptions of Bariatric Surgery among Diabetologists and Internists in Poland"

_jcm, 2022, doi:10.3390/jcm11072028_

Round 1

Reviewer 1 Report

This is an interesting and well written paper about knowledge and perceptions of bariatric surgery among internists and diabetologists. Because the literature on this topic is scarce, the manuscript could be suitable for the readers of JCM, especially in a monography about obesity surgery. 

Several limitations of the study should be more extensively discussed and and enhanced: 

  1. The number of the series is short. How this number compares with other research on this topic?. 
  2. It would be interesting to know the response rate. If the survey response rate is low, the generalizability and  validity of the results could be questioned. 
  3. The questionnaire used in the study was a not validated  one. Please discus this issue.  
  4. It would be interesting to know where the responders came from, since if they came from just one hospital, university, region or country, the generalizability of the results to other systems, countries, …, could be affected. 
  5. The title must reflect that the results are just form Poland, if this is the case. “Current Knowledge and perceptions of………and internists in Poland” 
  6. Supplementary Material: Question number 3:  the internist only option is missing. Check this please. 

Reviewer 2 Report

The premises and the focus of this paper are really interesting, but sample size is one of the more consistent limitations.

In materials and methods  it is reported that study population was recruited among physicians attending "2 large virtual Conferences..." It should be better reported how many were the physicians attending those event, given the low number of responders, and discussing this later on. It should also be clarified why so many (?) declined the survey. 

Round 2

Reviewer 2 Report

no further comments